# Efficiency Enhancement Strategies for Stable Bismuth-Based Perovskite and Its Bioimaging Applications

**DOI:** 10.3390/ijms24054711

**Published:** 2023-03-01

**Authors:** Liangyan Xiao, Linwei Huang, Weihaojia Su, Tianjun Wang, Haiying Liu, Zhongchao Wei, Haihua Fan

**Affiliations:** 1Guangdong Provincial Key Laboratory of Nanophotonic Functional Materials and Devices, School of Information and Optoelectronic Science and Engineering, South China Normal University, Guangzhou 510006, China; 2School of Biological Science and Medical Engineering, Southeast University, Nanjing 210096, China

**Keywords:** bismuth-based perovskite, quantum yield, rare earth, biocompatibility, bioimaging

## Abstract

Lead-free perovskite is one of the ideal solutions for the toxicity and instability of lead halide perovskite quantum dots. As the most ideal lead-free perovskite at present, bismuth-based perovskite quantum dots still have the problem of a low photoluminescence quantum yield, and its biocompatibility also needs to be explored. In this paper, Ce^3+^ ions were successfully introduced into the Cs_3_Bi_2_Cl_9_ lattice using a modified antisolvent method. The photoluminescence quantum yield of Cs_3_Bi_2_Cl_9_:Ce is up to 22.12%, which is 71% higher than that of undoped Cs_3_Bi_2_Cl_9_. The two quantum dots show high water-soluble stability and good biocompatibility. Under the excitation of a 750 nm femtosecond laser, high-intensity up-conversion fluorescence images of human liver hepatocellular carcinoma cells cultured with the quantum dots were obtained, and the fluorescence of the two quantum dots was observed in the image of the nucleus. The fluorescence intensity of cells cultured with Cs_3_Bi_2_Cl_9_:Ce was 3.20 times of that of the control group and 4.54 times of the control group for the fluorescence intensity of the nucleus, respectively. This paper provides a new strategy to develop the biocompatibility and water stability of perovskite and expands the application of perovskite in the field.

## 1. Introduction

In recent years, perovskite quantum dots (PQDs) have become one of the candidate materials in the new optoelectronic industry due to their excellent properties. Compared with traditional quantum dots, such as CdSe/CdS, the synthesis conditions and post-processing of PQDs are not harsh, and good optoelectronic properties can be obtained without relying on surface passivation. The main reason is the special defect tolerance [1] brought by the electronic structure—the band gap is opened between the two sets of antibonding orbitals. The vacancy defects easily fall within the conduction band and the valence band instead of the band gap, which makes PQDs less likely to generate deep defects and able to obtain a higher photoluminescence quantum yield (PLQY). At the same time, PQDs also have excellent characteristics, such as a simple synthesis process and large absorption cross-section [2]. These advantages mean that PQDs have great development prospects in the fields of semiconductor light emission, photovoltaics, photocatalysis, and photoelectric detection [3,4]. However, the instability of lead halide perovskite quantum dots (ultraviolet light [5,6,7,8,9], water [10,11], air [12,13,14], heat [15,16], etc.) and the more lethal lead toxicity severely limit their application and industrialization, especially in the field of biotechnology.

In order to improve the stability and reduce the toxicity of perovskites [17,18], the strategy of finding suitable low-toxicity elements to replace Pb to form lead-free perovskites has gradually occupied researchers’ attention. Among the numerous lead-free perovskite quantum dots [19,20,21], bismuth-based perovskite quantum dots (BQDs) are the most ideal lead halide perovskite substitutes due to the following characteristics. (1) They have a similar electronic structure to PQDs, so they have defect tolerance [22,23]. Deep defects are less likely to be generated and higher PLQY may be obtained. (2) They have low toxicity and excellent stability [24,25,26], especially high water resistance [27], making them completely different from other lead-free perovskites, such as Sb-based ones. (3) With larger exciton binding energy [28], the probability of radiative recombination increases. (4) They can be obtained by a simple solution synthesis process. Unlike Sn-based perovskites and double perovskites, they are not limited by high-temperature, oxygen-free, and other environments. However, the performance of BQDs is not as perfect as envisioned; due to the complex luminescence mechanism, there are still problems, such as a generally low PLQY, that need to be solved urgently. There is also a lack of research on the biological compatibility of BQDs. Currently, there are few reports on improving the PLQY of lead-free perovskites.

In this study, the rare earth ion Ce^3+^ was successfully doped into the bismuth-based perovskite Cs_3_Bi_2_Cl_9_ quantum dots by using an improved antisolvent method. The synthesized quantum dots (QDs) have high crystallinity and a small particle size distribution, and the PLQY of Cs_3_Bi_2_Cl_9_:Ce is increased from 12.88% to 22.12% compared with Cs_3_Bi_2_Cl_9_. Time-resolved fluorescence spectra show that the fluorescence lifetime of the quantum dots is decreased from 1.401 ns to 0.665 ns after doping Ce^3+^. The low-temperature fluorescence spectra reveal that the samples actually exhibit multi-peak emission. After doping with Ce^3+^, the fluorescence peak intensity of quantum dots at 410 nm, 433 nm, and 463 nm increases and the full width at half maximum (FWHM) narrows. The samples exhibit excellent biocompatibility and stability. After the quantum dots are cultured with human liver hepatocellular carcinoma (HepG2) cells for 24 h, a high-intensity upconversion fluorescence image is observed. The fluorescence intensity of cells cultured with Cs_3_Bi_2_Cl_9_:Ce is 3.20 times that of the control group and 4.54 times that of the control group for the fluorescence intensity of the nucleus, respectively. This work researches the upconversion fluorescence bioimaging of lead-free perovskites for the first time. It provides a new strategy to develop the biocompatibility and water stability of perovskite and expands the application of perovskite in the field of biotechnology.

## 2. Results and Discussion

### 2.1. Optimization of BQDs and Structural Characterization

According to previous reports [29,30], doping with new elements is an effective strategy to improve the PLQY of perovskite. Changing the energy band structure can introduce luminescent centers or affect electronic transitions to reduce non-radiative transitions. Doping rare earth ions can affect the band structures of perovskites. Therefore, Ce^3+^ ions are considered to be introduced to improve the optical properties. Both Cs_3_Bi_2_Cl_9_ and Cs_3_Bi_2_Cl_9_:Ce QDs were synthesized by a modified ligand-assisted resolvent method. By adjusting the type and dosage of precursor solvent, doping concentration, precursor ultrasonic time, temperature, and other process parameters, the highest PLQY of the samples was obtained (Appendix A). The doping concentration of CeCl_3_ ranged from 10 mol% to 45 mol%, and the optimal doping concentration was 30 mol%. This means that the ratio of the molar mass of Ce^3+^ used in the experimental synthesis to the total molar mass of (Ce^3+^ + Bi^3+^) was 0.3—that is, n (CeCl_3_)/n (CeCl_3_ + BiCl_3_) was 0.3. On the other hand, Cs_3_Bi_2_Br_9_ was also synthesized and compared with Cs_3_Bi_2_Cl_9_, and its PLQY was lower than that of Cs_3_Bi_2_Cl_9_, at only 8.2% (Appendix A).

Figure 1a,b show the absorption and photoluminescence (PL) spectra of the undoped Cs_3_Bi_2_Cl_9_ QDs and different Ce^3+^ doping concentration Cs_3_Bi_2_Cl_9_ QDs. Since the PLQY is the highest when the Ce^3+^ doping concentration is 30 mol%, the sample is abbreviated as Cs_3_Bi_2_Cl_9_:Ce 30% QDs. The absorption edge of Cs_3_Bi_2_Cl_9_ QDs is at 300 nm. The peak position is red-shifted after doping. The transmission spectrum of Cs_3_Bi_2_Cl_9_:Ce (30%) QDs has an obvious band-edge exciton peak at 325 nm. The band gap of Cs_3_Bi_2_Cl_9_ QDs is nearly 4.2 eV, which is increased compared to the 2.32 eV calculated for bulk Cs_3_Bi_2_Cl_9_ [31]; such a large band gap is one of the reasons for the blue light emission and low PLQY. Meanwhile, the band gap of Cs_3_Bi_2_Cl_9_:Ce (30%) QDs is approximately 3.8 eV, which is smaller than that of the undoped one. The incorporation of CeCl_3_ did not cause the emission spectrum and excitation spectrum of the Cs_3_Bi_2_Cl_9_:Ce (30%) QDs to change compared with Cs_3_Bi_2_Cl_9_ QDs. The possibility of Ce^3+^ ion emission can be ruled out considering the energy band structure of Ce^3+^ ion. The emission peak is located at 442 nm, with two shoulders at 420 and 466, respectively. The fluorescence profile of undoped QDs can be divided into four Gaussian peaks centered at 414, 434, 457, and 465 nm. Zhang et al. [32]. have observed a similar experimental phenomenon; their research shows that Cs_3_Bi_2_Cl_9_ QDs have both direct band gaps and indirect band gaps. As shown in Table 1, with the increase in the Ce^3+^ doping concentration, the PLQY of the samples first decreased, then increased, and then decreased. When the doping concentration reached 30 mol%, the PLQY reached a maximum of 22.12%.

The crystal structure information of the quantum dots before and after doping with CeCl_3_ was investigated by powder XRD characterization (Appendix A). The results show that the experimental values correspond well with the characteristic peaks of the standard card of Cs_3_Bi_2_Cl_9_ (PDF#46-1077), with a slight shift to a larger angle of the diffraction peak observed with increasing Ce^3+^ ion doping concentrations, which is attributed to the decrease in the lattice constant. With the increase in the Ce^3+^ ion doping concentration (from bottom to top), the characteristic peaks shown in the box formed by the dotted line in the figure gradually weakened. When the doping concentration reached as high as 45 mol%, the X-ray diffraction peaks were too weak to be recognized. The characteristic peak of the samples was sharper at 2Theta = 23.180°, especially when the doping concentration was 30 mol%, where the diffraction pattern presented a sharp main peak accompanied by many split small peaks. In addition, when the doping concentration was 30 mol%, the characteristic peak at 2Theta = 33.077° could be clearly observed, but it was difficult to demarcate this peak under other doping conditions. The results show that when the doping concentration is 30 mol%, the crystal structure of the samples is relatively stable. Under other conditions, the samples do not easily form a stable crystal structure, which may affect the luminescence properties of the samples. Combined with the fluorescence parameters of the samples, the change in PLQY may be attributed to the fact that Bi-Cl bonding and the crystal structure have a greater impact on the luminescence. While the Ce^3+^ doping concentration range is 10–20 mol%, the crystal structure may not be stable; when the doping concentration is 30 mol%, the crystal structure is stable, and Ce^3+^ can exert a benign effect on the Bi-Cl bond and accelerate radiation recombination; when the Ce^3+^ doping concentration is higher than 30 mol%, the large reduction in the core element Bi related to luminescence will result in a decrease in the PLQY.

Since the PLQY of Cs_3_Bi_2_Cl_9_:Ce 30% QDs is high at 22.12%, which is the highest among the doping samples, a series of research works were carried out for Cs_3_Bi_2_Cl_9_:Ce 30% QDs. First, the morphologies and structures of Cs_3_Bi_2_Cl_9_ and Cs_3_Bi_2_Cl_9_:Ce 30% QDs were characterized. In Figure 2a,b, the high-resolution TEM images show that the synthesized Cs_3_Bi_2_Cl_9_ and Cs_3_Bi_2_Cl_9_:Ce 30% QDs have good dispersion, a uniform particle size distribution, and small size, with an average size close to 2.7 nm (Figure 2c). Such a small size means that the samples have an obvious quantum size effect, which corresponds to the blue light emission and high PLQY of the material. The nanoparticles have a spherical morphology and show good crystallinity; the lattice fringes shown in Figure 2a,b correspond to the (1, 3, 10) plane of Cs_3_Bi_2_Cl_9_ and Cs_3_Bi_2_Cl_9_:Ce 30% QDs; since the radius of Ce^3+^ (103.4 pm) is smaller than the radius of Bi^3+^ (108 pm), the interplanar spacing of the crystal was slightly reduced after doping CeCl_3_. Affected by oleic acid, the electron diffraction pattern is difficult to be observed clearly (Appendix A). However, it can still be observed that both the diffraction patterns of Cs_3_Bi_2_Cl_9_ and Cs_3_Bi_2_Cl_9_:Ce 30% QDs are polycrystalline concentric rings, which also confirms their high crystallinity. In order to further study the internal structure changes of quantum dots before and after doping, the Raman spectra of the two quantum dots were also measured. As shown in Figure 2d, the wavenumbers of 235 cm^−1^, 262 cm^−1^, and 284 cm^−1^ can be designated as the characteristic peaks of Bi-Cl. The line width and peak position of the characteristic peak at 284 cm^−1^ changed significantly after doping Ce^3+^, and the replacement of the Ce^3+^ ion with the original Bi^3+^ ion in the crystal affected the coupling of Bi-Cl bonds, which resulted in a change in the Raman spectrum. In addition, compared with the undoped result, the Cs_3_Bi_2_Cl_9_:Ce 30% has a Ce-Cl characteristic peak at 180 cm^−1^, which proves the structural integrity of the sample and the successful doping of Ce^3+^.

As the optical properties of the material were affected by the surface area and the surface chemistry of the quantum dots, X-ray photoelectron spectroscopy was carried out to evaluate the surface structure and state of Cs_3_Bi_2_Cl_9_ and Cs_3_Bi_2_Cl_9_:Ce 30% QDs. As shown in Figure 3a–c, peaks of Cs, Bi, and Cl appeared for both undoped and doped CeCl_3_ nanoparticles. The fitted XPS peaks of Cs-3d correspond to the 3d-5/2 and 3d-3/2 bonds of Cs+, respectively (Figure 3a); the peaks of Cl-3d correspond to the 3d-5/2 and 3d-3/2 bonds of Cl-, respectively (Figure 3b); and the peaks of Bi-4f are attributed to 4f-7/2 and 4f-5/2 bonding of Bi^3+^ (Figure 3c), respectively. Compared with the XPS peaks of Cs_3_Bi_2_Cl_9_ QDs (4f-7/2, 159.4 eV), the Bi-4f binding energy of Cs_3_Bi_2_Cl_9_:Ce 30% QDs (159.6 eV) is slightly increased, which is due to the introduction of Ce^3+^ as an electron accepter. In Figure 3d, Cs_3_Bi_2_Cl_9_:Ce 30% QDs show two additional peaks at 885 eV and 903 eV, which are attributed to the Ce-3d signal of Ce^3+^ ion. These two peaks prove that Ce^3+^ ions have been successfully doped into the nanoparticles, and it is noted that there is a small amount of Ce4+ signal in the sample due to the oxidation of Ce^3+^. In addition, quantitative analysis by XPS (Appendix A) showed that the surface Cl/Bi atomic ratios of the undoped and doped samples were 1.5 and 2.91, respectively, which were lower than the ideal atomic ratio. These results indicated that the surface of the sample was still in a Cl-poor state; there were many dangling bonds and defects on the surfaces of the doped and undoped Cs_3_Bi_2_Cl_9_ QDs. The surface Ce/(Ce + Bi) ratio of Cs_3_Bi_2_Cl_9_:Ce 30% QDs is 0.3; Inductively Coupled Plasma Mass Spectrometry (ICP) showed the actual doping concentration of Ce^3+^ and Bi^3+^ ions inside the crystal. From Table 2, it can be seen that the ratio of Ce/(Ce + Bi) is close to the synthesis design ratio. When the experimental doping concentration is 30 mol%, the molar concentration ratio of Ce^3+^ and Ce^3+^ + Bi^3+^ inside the crystal is 31.07%. Combining the experimental results of XRD and XPS, the chemical formulas of the synthesized samples can be determined as Cs_3_Bi_2_Cl_9_ and Cs_3_(Bi_1.4_Ce_0.6_)Cl_9_, respectively.

### 2.2. Optical Properties

Poor stability and toxicity have been the main challenges hindering the development of perovskites, so the stability of perovskites was investigated. Two samples were synthesized and placed in the air for 30 days under non-dark conditions; at different time points, the fluorescence intensities of the samples excited by 375 nm light were measured. The normalized results are shown in Figure 4a. After 7 days, the fluorescence intensity of the samples was more than 90% of the initial intensity. After 14 days, the fluorescence intensity of Cs_3_Bi_2_Cl_9_ QDs was 83% of the initial value, and the fluorescence intensity of Cs_3_Bi_2_Cl_9_:Ce 30% QDs was 86% of the initial value. After 30 days, the fluorescence intensity of the samples remained above 70% of the initial intensity. At the same time, the water stability of the samples was also investigated (Figure 4b). Here, 2 mL of the colloidal solution to be tested was mixed with 1 mL of deionized water. The fluorescence spectra of the samples were measured at different time points. Consistent with a previous report [27], water molecules react with BiCl_3_, and the generated BiOCl coats the surface of the quantum dots to prevent the perovskite from being damaged. Due to the surface passivation of BiOCl, the samples have good water stability. The fluorescence intensity of the samples was even higher than the initial value in the first 6 h, and the intensity could still be maintained at 79% and 76% of the original intensity after 12 h, respectively.

To explore the mechanism of the PLQY enhancement of the samples after doping with Ce^3+^, the time-resolved photoluminescence spectrum has been measured to reveal the exciton recombination dynamics of Cs_3_Bi_2_Cl_9_ QDs. Appendix A show the fluorescence decay of Cs_3_Bi_2_Cl_9_ and Cs_3_Bi_2_Cl_9_:Ce 30% QDs, and the fluorescence lifetimes were fitted by bi-exponential curves. The lifetimes of Cs_3_Bi_2_Cl_9_ QDs were fitted with a short fluorescence lifetime of 1.401 ns (τ1) accounting for 70.65% and a long fluorescence lifetime of 10.394 ns (τ2) accounting for 29.35%, and the lifetimes of Cs_3_Bi_2_Cl_9_:Ce 30% QDs were fitted with a short fluorescence lifetime (τ1) of 0.665 ns accounting for 82.28% and the long fluorescence lifetime (τ2) of 3.419 ns accounted for 17.72%. Such a short fluorescence lifetime verifies that the doping of Ce^3+^ can accelerate the radiative recombination process of quantum dots.

Due to the limited thermal vibration of the lattice at low temperatures, the probability of the non-radiative recombination of the material is reduced, and the low-temperature photoluminescence spectrum of the sample was measured to explore the intrinsic exciton emission properties of the material. As shown in Figure 4b, the spectra of the samples show four clear and distinct luminescence peaks at 150 K, located at 410 nm, 433 nm, 463 nm, and 466 nm, respectively. As shown in Table 3 and Table 4, after doping, the fluorescence intensities of Peak1, Peak2, and Peak3 increased from 38.7, 64.1, and 27.1 to 47.9, 74.6, and 42.4, respectively, and the FHWM of Peak1 and Peak2 decreased from 17.9 nm and 29.8 nm to 14.1 nm and 24.0 nm, respectively. Appendix A shows the multipeak fitting of the samples’ room-temperature photoluminescence profiles. Appendix A list the parameters of the four fitting peaks. It can be seen from Table 3, Table 4, Appendix A that with the increase in temperature, the FHWM of Peak1 and Peak2 changes very little and the intensity decreases. The intensity of Peak3 and Peak4 increases, and the FHWM of Peak3 becomes wider. This result is very similar to references [32,33]. The emission of Peak1 and Peak2 may be related to exciton recombination, which may be attributed to the direct transition of the sample. Peak3 and Peak4 may be related to recombination with phonon participation, which can be attributed to self-trapped excitons or indirect transition bands. The incorporation of Ce^3+^ may be beneficial to the exciton recombination process, which is consistent with the conclusions obtained from the fluorescence lifetime experiments.

The upconversion properties of materials are of great significance in biological imaging, so the upconversion fluorescence of the two samples was measured. Cs_3_Bi_2_Cl_9_ and Cs_3_Bi_2_Cl_9_:Ce 30% QDs were excited by a 750 nm femtosecond pulsed laser, and the fluorescence was detected; see the Appendix A for the detailed up-conversion fluorescence characteristics of the samples. The relationship between the fluorescence intensity and the excitation light power was obtained, and the fitted slopes were 1.74 and 1.72, respectively. In addition, Rhodamine-6G was used as the reference, and the upconversion absorption cross-sections of the samples were obtained.

To explore the interaction between nanoparticles and cells, the biocompatibility of the materials was also investigated. The standard MTT assay was adopted to evaluate the cellular toxicity of Cs_3_Bi_2_Cl_9_ and Cs_3_Bi_2_Cl_9_:Ce 30% QDs. HepG2 cells were co-incubated with different concentrations of nanomaterials for 24 h and the cell viability was measured. The results are shown in Figure 5. At all concentrations, the cell viability of the Cs_3_Bi_2_Cl_9_:Ce 30% QD group was higher than that of the Cs_3_Bi_2_Cl_9_ QD group. When the concentrations of Cs_3_Bi_2_Cl_9_ and Cs_3_Bi_2_Cl_9_:Ce 30% QDs are low at 10 μg/mL, the viability of HepG2 cells is high at 95.2% and 98.2%, respectively. When the concentration increases to 40 μg/mL, the viability of the Cs_3_Bi_2_Cl_9_ QD group decreases to 92%,while the result for the Cs_3_Bi_2_Cl_9_:Ce 30% QD group is 96.1%. With the increase in concentration, the data gap between the two experimental groups widened. When the concentration is as high as 200 μg/mL, the viability of the Cs_3_Bi_2_Cl_9_ QD group is 81.3%, and that for the Cs_3_Bi_2_Cl_9_ QD group is 91%. Doping Ce not only improves the PLQY, but also reduces the toxicity of quantum dots. Compared with the encapsulated lead halide perovskites CsPbBr3@PMMA previously reported [34], when the cell viability was at 90%, the concentration of PQDs was 60 μg/mL, the concentration of Cs_3_Bi_2_Cl_9_ QDs was 85 μg/mL, and the concentration of Cs_3_Bi_2_Cl_9_:Ce 30% QDs was 200 μg/mL. In this work, the concentration of Cs_3_Bi_2_Cl_9_ QDs was approximately 1.42 times that of PQDs, and that of Cs_3_Bi_2_Cl_9_:Ce 30% QDs was approximately 3.33 times that of PQDs.

On the other hand, the cellular uptake of the samples was quantitatively determined by the ICP-MS assay. The results are shown in Table 5. The uptake of Cs_3_Bi_2_Cl_9_ QDs by a single HepG2 cell was 4.42 × 10^−5^ μg/cell, and the uptake of Cs_3_Bi_2_Cl_9_:Ce 30% QDs was 6.33 × 10^−5^ μg/cell, which is close to the uptake of carbon quantum dots. The results show that bismuth-based perovskites are more easily absorbed by cells, and the doping of Ce^3+^ can enhance the biocompatibility.

Such low toxicity, high stability, and high PLQY make the material an excellent candidate for fluorescent probes, so upconversion imaging experiments were performed. The dependence of the upconversion fluorescence intensity of the two QDs was measured and is shown in Appendix A. After incubating HepG2 cells with two samples at a concentration of 50 μg/mL for 24 h, the HepG2 cells were washed three times with PBS. Then, the HepG2 cells were scanned by a laser scanning microscope excited with a 750 nm femtosecond pulsed laser, and the upconversion fluorescence images were captured, as shown in Figure 6. The results showed that the cells in the control group (captured without QDs) and the experimental group (captured with QDs) were both in good condition. Figure 6a–c show the brightfield and fluorescence images of HepG2 cells in the control group, Figure 6d–f show the brightfield and fluorescence images of HepG2 cells incubated with Cs_3_Bi_2_Cl_9_ QDs for 24 h, and Figure 6g–i show the brightfield and fluorescence imaging of HepG2 cells incubated with Cs_3_Bi_2_Cl_9_:Ce 30% QDs for 24 h. In the blue and green channels, the cells of the experimental group showed strong upconversion fluorescence emission, the cell outline and internal structure could be clearly observed, and the fluorescence of cells incubated with Cs_3_Bi_2_Cl_9_:Ce 30% QDs was obviously stronger.

In order to quantitatively analyze the enhancement effect of Cs_3_Bi_2_Cl_9_ and Cs_3_Bi_2_Cl_9_:Ce 30% QDs on cell imaging, the fluorescence intensities of single cells (outlined in red) have been measured. The results are shown in Table 6. Affected by the luminescence of quantum dots, the fluorescence intensities of the green and blue channels of the cells in the experimental group were significantly higher than those of the control group. In the green channel, the maximum fluorescence intensity of cells in the control group, the Cs_3_Bi_2_Cl_9_ QD group, and the Cs_3_Bi_2_Cl_9_:Ce 30% QD group was 48, 76, and 81, respectively. In the blue channel, the maximum fluorescence intensity of cells in the control group, the Cs_3_Bi_2_Cl_9_ QD group, and the Cs_3_Bi_2_Cl_9_:Ce 30% QD group was 44, 104, and 141, respectively. In this channel, the maximum fluorescence intensity of cells in the Cs_3_Bi_2_Cl_9_ and Cs_3_Bi_2_Cl_9_:Ce 30% QD group was around 2.36 times and 3.20 times that of the control group, respectively. The results indicated that the bismuth-based perovskite quantum dots have good potential for upconversion imaging, and the doping of Ce^3+^ could improve the upconversion imaging effect.

Another surprising result was that the intense fluorescence of some quantum dots in the nucleus was observed. As shown in Table 7, the luminescence intensity of the quantum dots in the nucleus (outlined by white circle) was also measured. The minimum fluorescence intensities of the nuclei in these groups were similar. In the blue channel, the maximum fluorescence intensity of the nuclei in the control group, Cs_3_Bi_2_Cl_9_ QD group, and Cs_3_Bi_2_Cl_9_:Ce 30% QD group was 28, 92, and 127, respectively. In the control group, the maximum fluorescence intensity of nuclei was significantly lower. However, the maximum fluorescence intensity of nuclei in the experimental group was close to the maximum fluorescence intensity of cells. This result showed that (1) the nucleus in the control group had almost no luminescence from quantum dots; (2) in the blue channel, the maximum fluorescence intensity of the nucleus in the Cs_3_Bi_2_Cl_9_ QD group and Cs_3_Bi_2_Cl_9_:Ce 30% QD group was around 3.29 times and 4.54 times, respectively. The nuclear pore complex as a nuclear-cytoplasmic exchange channel can be regarded as a special transmembrane transport protein. For substances with diameters below 10 nm, such as ions and small molecules, passive diffusion can be used to pass through the hydrophilic channel in principle. Some related studies have shown that a positively charged surface and ultra-small size are important factors for nanomaterials to enter the nucleus [35]. The zeta potential results show that the surfaces of the two quantum dots are positively charged (Appendix A). The nucleophilicity of quantum dots might due to the small particle size, positively charged surfaces of nanoparticles, and their stable and hydrophilic properties in cells. In conclusion, Cs_3_Bi_2_Cl_9_ and Cs_3_Bi_2_Cl_9_:Ce 30% QDs already have extremely high biocompatibility, and the rare nucleophilicity even indicates that the samples have potential for targeted applications.

## 3. Materials and Methods

### 3.1. Synthesis of Cs_3_Bi_2_Cl_9_ QDs and CeCl_3_-Doped Cs_3_Bi_2_Cl_9_ QDs

Both Cs_3_Bi_2_Cl_9_ QDs (BQDs) and CeCl_3_-doped Cs_3_Bi_2_Cl_9_ QDs ((Ce^3+^)BQDs) were synthesized by an improved ligand-assisted resolvent method. See the Appendix A for the detailed experimental reagents and process.

### 3.2. Structural Characterization

The structural components of the synthesized quantum dots were characterized by transmission electron microscopy, X-ray powder diffraction, and other characterization methods. See the Appendix A for the detailed experimental equipment and steps.

### 3.3. Optical Characterization

The optical properties of the samples were obtained using a UV–Vis spectrophotometer, fluorescence spectrophotometer, etc. The photoluminescence quantum yield (PLQY) of the samples was obtained by an indirect method. See the Appendix A for the detailed experimental equipment, process, and principles. The fluorescence lifetime and low-temperature fluorescence spectra of quantum dots were tested by a transient fluorescence spectrometer (FLS-920, Edinburgh, UK) with an excitation wavelength of 365 nm. The upconversion fluorescence characteristics of quantum dots were investigated using a 76 MHz femtosecond pulsed laser with an excitation wavelength of 750 nm. See the Appendix A for the details.

### 3.4. Cell Viability and Cellular Uptake

The human liver hepatocellular carcinoma (HepG2) cells used in the laboratory were obtained from the Cell Laboratory of the Cell Resource Center of the Chinese Academy of Sciences. The relative cell viability of the samples was determined by the MTT method, and the uptake of quantum dots by individual cells was determined using Inductively Coupled Plasma Mass Spectrometry (ICP-MS) (ICAP-qc, Thermo Fisher, Dreieich, Germany). See the Appendix A for the detailed experimental process.

### 3.5. Cell Imaging

Upconversion fluorescence images of HepG2 cells were obtained using a multiphoton laser scanning microscope (FV1200MPE, Olympus, Japan) with an excitation wavelength of 750 nm. See the Appendix A for the detailed measurement method.

## 4. Conclusions

In conclusion, a CeCl_3_-doped bismuth-based perovskite quantum dot is reported with a PLQY of up to 22.12% when the doping concentration is 30 mol%. The synthesized nanocrystals have a small particle size and high crystallinity. The chemical formulas of the samples are Cs_3_Bi_2_Cl_9_ and Cs_3_(Bi_1.4_Ce_0.6_)Cl_9_, respectively. The quantum dots doped with Ce^3+^ have a shorter fluorescence lifetime and a significantly higher sub-peak fluorescence intensity. Our samples exhibit excellent stability and excellent biocompatibility. Nonlinear optical imaging of BQDs has been studied for the first time, and impressive upconversion fluorescence imaging effects and rare nuclear affinity have been observed under femtosecond pulsed light at 750 nm. Our work expands the development prospects of BQDs in the field of optoelectronics.

## Figures and Tables

**Figure 1 ijms-24-04711-f001:**
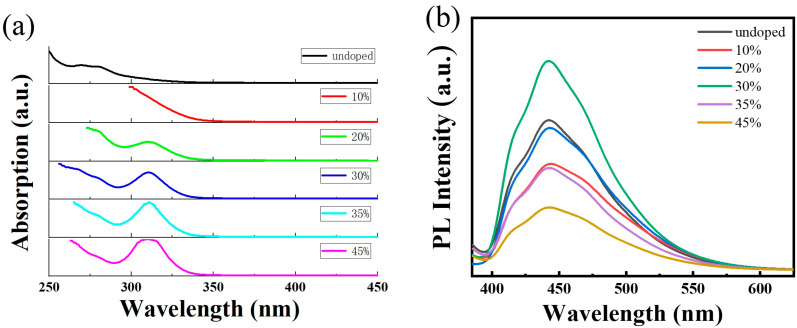
(**a**) Transmission spectrum of Cs_3_Bi_2_Cl_9_ QDs and Cs_3_Bi_2_Cl_9_:Ce QDs; (**b**) photoluminescence spectrum of Cs_3_Bi_2_Cl_9_ QDs and Cs_3_Bi_2_Cl_9_:Ce QDs with excitation wavelength of 375 nm.

**Figure 2 ijms-24-04711-f002:**
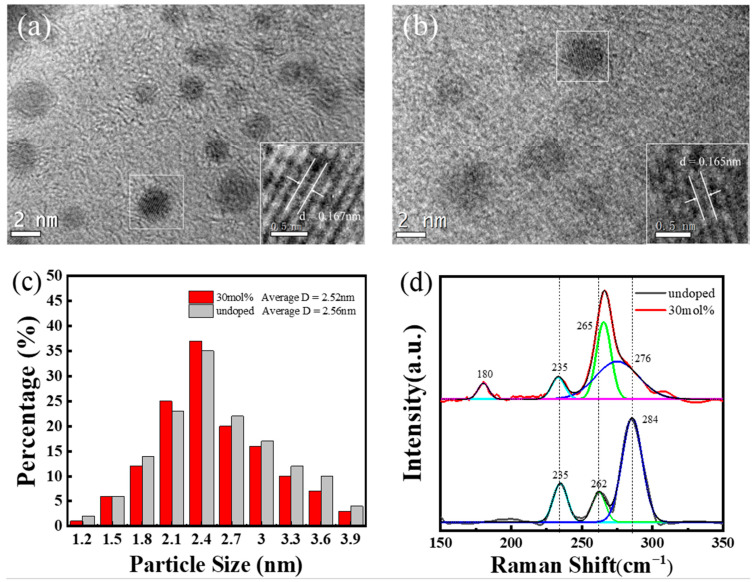
(**a**) High-resolution TEM image of Cs_3_Bi_2_Cl_9_ QDs; (**b**) high-resolution TEM image of Cs_3_Bi_2_Cl_9_:Ce 30% QDs; (**c**) particle size distribution of Cs_3_Bi_2_Cl_9_ and Cs_3_Bi_2_Cl_9_:Ce 30% QDs; (**d**) Raman spectra of Cs_3_Bi_2_Cl_9_ and Cs_3_Bi_2_Cl_9_:Ce 30% QDs. The upper part: the red line is the experimental data of the sample doped with 30% Ce, and the other color lines are single-peak fitting, The lower part: the black line is the experimental data of undoped samples, and the other color lines are single-peak fitting.

**Figure 3 ijms-24-04711-f003:**
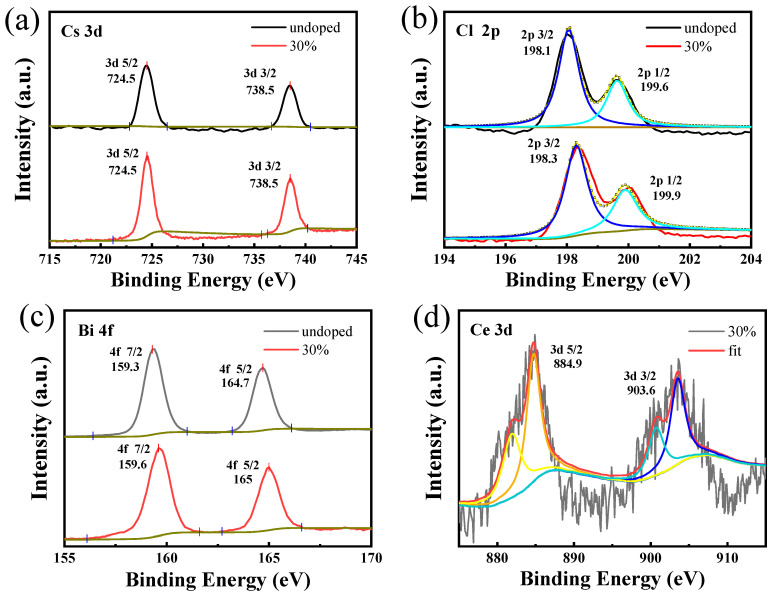
XPS spectra of Cs_3_Bi_2_Cl_9_ and Cs_3_Bi_2_Cl_9_:Ce 30% QDs. (**a**) Energy spectrum analysis of Cs element; (**b**) energy spectrum analysis of Cl element; The black line is the experimental data, Dark blue and sky-blue lines are single-peak fitting (**c**) energy spectrum analysis of Bi element; (**d**) energy spectrum analysis of Ce element. The black line is the experimental data, the red line is the fitting of the whole spectrum, and the other color lines are single-peak fitting.

**Figure 4 ijms-24-04711-f004:**
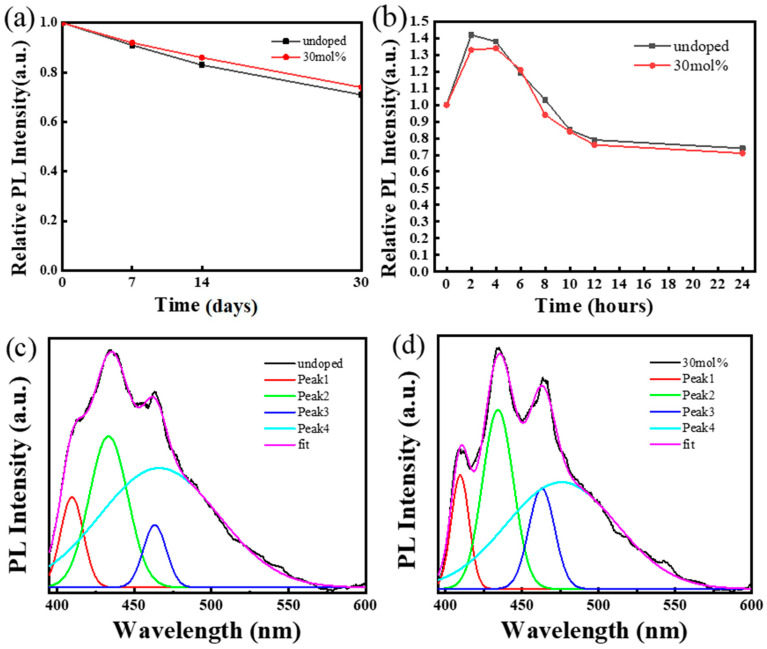
(**a**) Air stability of samples under non-dark conditions, results normalized; (**b**) water stability experiments of Cs_3_Bi_2_Cl_9_ and Cs_3_Bi_2_Cl_9_:Ce 30% QDs and the relative PL intensities of samples in water as a function of time, results normalized; (**c**) low-temperature (150 K) photoluminescence spectrum and related fitting peaks of Cs_3_Bi_2_Cl_9_ QDs; (**d**) low-temperature (150 K) photoluminescence spectrum and related fitting peaks of Cs_3_Bi_2_Cl_9_:Ce 30% QDs.

**Figure 5 ijms-24-04711-f005:**
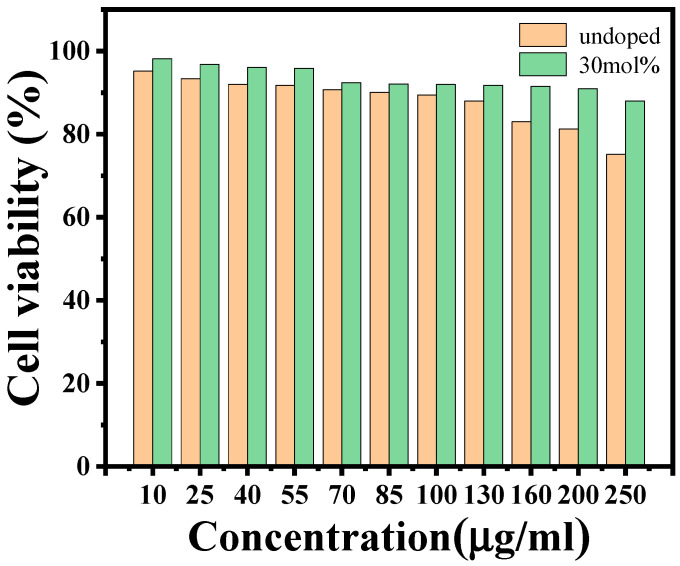
The effect of different concentrations of Cs_3_Bi_2_Cl_9_ and Cs_3_Bi_2_Cl_9_:Ce 30% QDs on the viability of HepG2 cells after co-culture with HepG2 cells for 24 h as determined by MTT assay.

**Figure 6 ijms-24-04711-f006:**
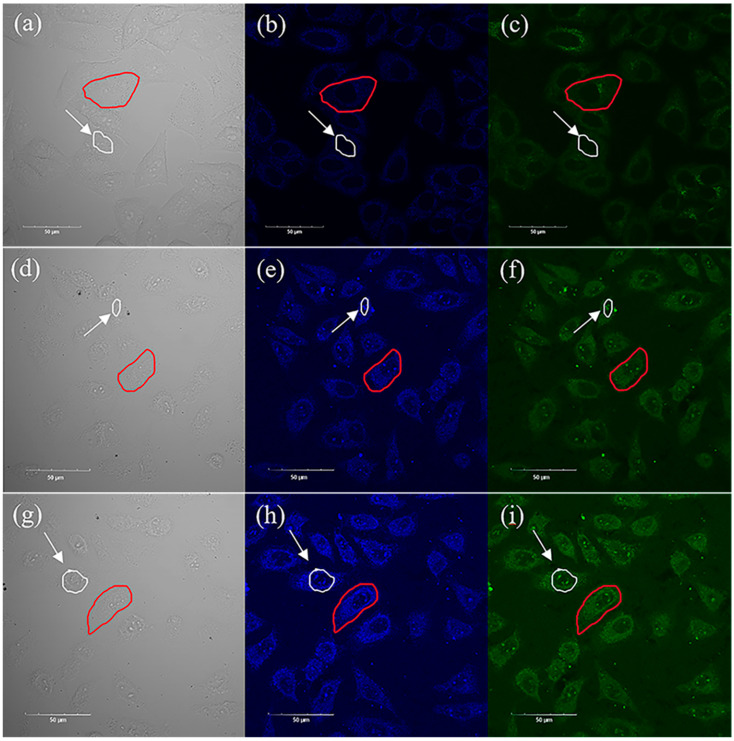
(**a**–**c**) Brightfield images and fluorescence images of HepG2 cells in the control group, where (**a**) is the brightfield image, (**b**) is the fluorescence image with the blue channel, (**c**) is the fluorescence image with the green channel. (**d**–**f**) Brightfield images and fluorescence imaging of HepG2 cells co-incubated with Cs_3_Bi_2_Cl_9_ QDs for 24 h, where (**d**) is the brightfield image, (**e**) is the fluorescence image with the blue channel, (**f**) is the fluorescence image with the green channel. (**g**–**i**) Brightfield and fluorescence imaging of HepG2 cells incubated with Cs_3_Bi_2_Cl_9_:Ce 30% QDs for 24 h, where (**g**) is the brightfield image, (**h**) is the fluorescence image with the blue channel, and (**i**) is the fluorescence image with the green channel. The red circles in the figure show the target cells to be measured, the white circles and the arrows show the target cell nuclei, and the scale of the icon is 50 μm.

**Table 1 ijms-24-04711-t001:** Fluorescence information for samples.

QDs	PLQY (%)	Band Gap (eV)
undoped	12.88	4.2
10 mol%	10.32	4.0
20 mol%	12.47	3.7
30 mol%	22.12	3.8
35 mol%	9.73	3.8
45 mol%	5.78	3.7

**Table 2 ijms-24-04711-t002:** Actual dopant concentrations obtained by Inductively Coupled Plasma Mass Spectrometry (ICP-MS).

	Undoped	10 mol%	20 mol%	30 mol%	45 mol%
^a^ Ce (wt %)		2.78	5.94	9.32	14.18
^b^ Bi (wt %)	65.28	40.63	37.38	30.82	22.56
^c^ Ce/(Bi + Ce) mol%		9.26	19.16	31.07	43.60

^a^ Ce accounts for the total mass fraction of the sample; ^b^ Bi accounts for the total mass fraction of the sample; ^c^ the moles of Ce account for the percentage of the total moles of (Bi + Ce).

**Table 3 ijms-24-04711-t003:** Low-temperature fluorescence parameters of fitted peaks for Cs_3_Bi_2_Cl_9_ QDs, results normalized.

	Peak1	Peak2	Peak3	Peak4
Position (nm)	410	433	463	466
Height	38.7	64.1	27.1	50.8
FHWM (nm)	17.9	29.8	18.3	92.1

**Table 4 ijms-24-04711-t004:** Low-temperature fluorescence parameters of fitted peaks for Cs_3_Bi_2_Cl_9_:Ce 30% QDs, results normalized.

	Peak1	Peak2	Peak3	Peak4
Position (nm)	410	433	463	476
Height	47.9	74.6	42.4	45.0
FHWM (nm)	14.1	24.0	20.4	87.0

**Table 5 ijms-24-04711-t005:** Uptake of Cs_3_Bi_2_Cl_9_ and Cs_3_Bi_2_Cl_9_:Ce 30% QDs by HepG2 cells.

QDs	Uptake (μg/Cell)
undoped	4.42 × 10^−5^
30 mol%	6.33 × 10^−5^

**Table 6 ijms-24-04711-t006:** Upconversion fluorescence intensity of selected cells.

Sample	(b) Control Blue	(c) Control Green	(e) Cs_3_Bi_2_Cl_9_ QDs Blue	(f) Cs_3_Bi_2_Cl_9_ QDs Green	(h) Cs_3_Bi_2_Cl_9_:Ce 30% QDs Blue	(i) Cs_3_Bi_2_Cl_9_:Ce 30% QDs Green
Intensity (min)	9	9	10	11	10	12
Intensity (max)	44	48	104	76	141	81
Intensity (mean)	19.51	24.60	30.45	28.25	37.11	30.32

**Table 7 ijms-24-04711-t007:** Upconversion fluorescence intensity of quantum dots in the nucleus.

Sample	(b) Control Blue	(c) Control Green	(e) Cs_3_Bi_2_Cl_9_ QDs Blue	(f) Cs_3_Bi_2_Cl_9_ QDs Green	(h) Cs_3_Bi_2_Cl_9_:Ce 30% QDs Blue	(i) Cs_3_Bi_2_Cl_9_:Ce 30% QDs Green
Intensity (min)	10	10	13	11	14	13
Intensity (max)	28	32	92	69	127	87
Intensity (mean)	15.11	17.35	27.33	29.88	36.95	27.71

## Data Availability

All data associated with this article are included in the article.

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
