# Peer review of "Efficiency Enhancement Strategies for Stable Bismuth-Based Perovskite and Its Bioimaging Applications"

_ijms, 2023, doi:10.3390/ijms24054711_

Round 1
Reviewer 1 Report
This work introduced bismuth-based perovskite’s high quantum yield and optical properties. This manuscript is well written and organized, may be accepted after minor revisions solving questions as follow.
1. Line 23, the sentence should be ’in the field of biotechnology’ or in the “biotechnology field”.
2. Line 111, authors claim that emission peak is at 446nm and the FWHM is nearly 70nm, and this can be attributed to self-trapped excitons. How did authors confirm this as authors also pointed out that there could be other peaks around 425 and 470 nm? If the peak around 446nm is decomposed into several different peaks, the FWHM at 446nm will not be 70nm.
3. Figure 4(c)(d), authors claim that there are four peaks for PL spectra. Can authors explain their physical mechanism? As cited papers claim there are 3 peaks. Zhang et al. [29].
Author Response
We would like to thank the comments and suggestions of the reviewer which are definitely helpful for improving the quality of our manuscript. The responses (abbreviated as R) to these comments (abbreviated as C) and the changes made in the revised manuscript (marked in red color) are described in detail in the following.
This work introduced bismuth-based perovskite’s high quantum yield and optical properties. This manuscript is well written and organized, may be accepted after minor revisions solving questions as follow.
C1: Line 23, the sentence should be ’in the field of biotechnology’ or in the “biotechnology field”.
R1: According to the suggestion of the reviewer, the sentence has been revised.
C2: Line 111, authors claim that emission peak is at 446nm and the FWHM is nearly 70nm, and this can be attributed to self-trapped excitons. How did authors confirm this as authors also pointed out that there could be other peaks around 425 and 470 nm? If the peak around 446nm is decomposed into several different peaks, the FWHM at 446nm will not be 70nm.
R2: Thanks for the reviewer's suggestion. Our expression in the manuscript is not clear. In fact, Figure 1b described by line111 is the fluorescence spectrum of the samples. We carefully checked the spectra, and the emission peak of the sample was located at 442nm. In this work, the luminescence band of both doped and undoped samples is wide. From the spectrum, we can see that there are two shoulders besides the main peak. This means that there are multiple emission peaks superimposed. It is not accurate to define the FWHM for 446nm. We have performed a multi-peak fitting for the spectrum. As shown in Figure S6, the peak position and half-height width of multiple peak fitting are shown in Table S3 and Table S4. Similar results have been reported in [1]. In reference [1], Zhang et al. attributed the narrow emission peak with decreasing intensity as the temperature increased to direct transitions. As the temperature rises, the broadband emission peak with the intensity rising is attributed to the phonon-assisted indirect transitions. Reference [2] summarizes that the characteristic of self-trapped excitons is broadband transmission,Narrowband emission is attributed to Free excitons. Since self-trapped excitons (STEs) formed due to the strong carrier–phonon interaction in these materials ,if the temperature decreases, the emission peak FWHM of self-trapped excitons decreases. This paper synthesizes the conclusions of the above two references. We change the sentences:“The emission peak is located at 446nm, and the FWHM is nearly 70nm. Such a broad luminescence peak can be attributed to self-trapped excitons. The emission band of ODs can be decomposed into several emission peaks. Except for the highest emission peak at 446nm, there are other peaks near 425nm and 470nm.” to “The emission peak is located at 442nm, with two shoulders at 420 and 466 respectively. the fluorescence profile of undoped QDs can be divided into four Gaussian peaks centered at 414, 434, 457and 465 nm.” Table S3 and Table S4and Figure S5 have been added in supporting information.
Figure S5 (a) Room-temperature photoluminescence spectrum and related fitting peaks of Cs3Bi2Cl9 QDs; (b) room-temperature photoluminescence spectrum and related fitting peaks of Cs3Bi2Cl9: Ce 30% QDs.
Table S3. Room-Temperature Fluorescence parameters of fitted peaks for Cs3Bi2Cl9 QDs, results normalized.
|
|
Peak1 |
Peak2 |
Peak3 |
Peak4 |
|
Position (nm) |
414 |
434 |
457 |
465 |
|
Height |
24.1 |
39.4 |
36.0 |
44.7 |
|
FHWM (nm) |
15.4 |
29.8 |
46.7 |
107.1 |
Table S4. Room-Temperature Fluorescence parameters of fitted peaks for Cs3Bi2Cl9: Ce 30% QDs, results normalized.
|
|
Peak1 |
Peak2 |
Peak3 |
Peak4 |
|
Position (nm) |
414 |
434 |
457 |
473 |
|
Height |
28.8 |
43.3 |
44.8 |
37.2 |
|
FHWM (nm) |
17.0 |
30.3 |
48.8 |
97.0 |
C3: Figure 4(c)(d), authors claim that there are four peaks for PL spectra. Can authors explain their physical mechanism? As cited papers claim there are 3 peaks. Zhang et al. [29].
R3: In this work the Photoluminescence profile of the samples can be divided into several Gaussian peaks. Similar results have been reported in [1]. In reference [1], Zhang et al. attributed the narrow emission peak with decreasing intensity as the temperature increased to direct transitions. As the temperature rises, the broadband emission peak with the intensity rising is attributed to the phonon-assisted indirect transitions. Reference [2] summarizes that the characteristic of self-trapped excitons is broadband transmission. Narrow band emission is attributed to Free excitons. Since self-trapped excitons (STEs) formed due to the strong carrier–phonon interaction in these materials, as the temperature decreases, the FWHM of emission peak due to self-trapped excitons decreases. This paper synthesizes the conclusions of the above two references. Theoretically, the radiative recombination of excitons through a direct bandgap transition decreases with increasing temperature because of thermal quenching. In stark contrast to this behavior, recombination through an indirect bandgap transition can be greatly enhanced by increasing the temperature because it requires additional momentum compensation from phonons to obey momentum conservation. Table 3 and Table 4 are the parameters of the four fitting peaks of the samples at low temperature, and Table S3 and S4 list the parameters of the four fitting peaks of the samples at room temperature. By comparing the four tables, we can get some conclusions: With the increase of temperature, the FHWM of peak 1 and peak 2 change very little and the intensity becomes weak. They can be attributed to the direct bandgap transition; With the increase of temperature, the intensity of peak 3 and peak 4 rise, which can be attributed to the indirect bandgap transition. With the increase of temperature, the FHWM of peak 3 increases. Peak 3 is probably attributed to direct bandgap transition.
We changed the sentences in line 264 to” Figure S6 is the multipeak fitting of the samples room temperature photoluminescence profile. Table S3 and S4 list the parameters of the four fitting peaks. It can be seen from Table 3, Table 4 ,Table S3 and S4: with the increase of temperature, the FHWM of peak 1 and peak 2 change very little and the intensity decreases. the intensity of peak 3 and peak 4 increases, the FHWM of peak 3 become wider. This result is very similar to references [32] and [34]. The emission of Peak1, Peak2, Peak3 may be related to exciton recombination, which may be attributed to the direct transition of the sample. Peak3 and Peak4 may be related to recombination with phonon participation, which can be attributed to self-trapped excitons or indirect transition bands.”
[1] Zhang, Y.; Yin, J.; Parida, M. R.; Ahmed, G. H.; Jun, P.; Bakr, O. M.; Brédas, J-L; Mohammed, O. F. Direct-indirect nature of the bandgap in lead-free perovskite nanocrystals. J. Phys. Chem. Lett, 2017, 8, 3173-3177.
[2] Kahmann, S.; Tekelenburg, E. K. ; Duim, H.; Kamminga, M. E.; Loi, M. A. Extrinsic nature of the broad photoluminescence in lead iodide-based Ruddlesden–Popper perovskites. Nat Comm. 2020, 11, 2344
|
|
Peak1 |
Peak2 |
Peak3 |
Peak4 |
|
Position (nm) |
410 |
433 |
463 |
466 |
|
Height |
38.7 |
64.1 |
27.1 |
50.8 |
|
FHWM (nm) |
17.9 |
29.8 |
18.3 |
92.1 |
Table 3. Low-Temperature Fluorescence parameters of fitted peaks for Cs3Bi2Cl9 QDs, results normalized.
|
|
Peak1 |
Peak2 |
Peak3 |
Peak4 |
|
Position (nm) |
410 |
433 |
463 |
476 |
|
Height |
47.9 |
74.6 |
42.4 |
45.0 |
|
FHWM (nm) |
14.1 |
24.0 |
20.4 |
87.0 |
Table 4. Low-Temperature Fluorescence parameters of fitted peaks for Cs3Bi2Cl9: Ce 30% QDs, results normalized.

Reviewer 2 Report
The authors reported Ce doped Cs3Bi2Cl9 quantum dots for upconversion fluorescence bioimaging. Halide-based perovskite is a hot area and this paper could be attractive to many readers. This work is well organized and performed. However, there still some flaws in the manuscript, including references and writing. Therefore, a major revision must be made before I can reconsider it for publication.
1. Many typos in the manuscript in the paper and Figures. For example, line 171 “235 cm-1, 262 cm-1 and 284 cm-1”; Figure 2c, I do not know what the B and C mean; Figure 2a and 2b, the number on the scale bar should be the same size.
2. Figure 2c, in principle, the particle size distribution should be Gaussian distribution, while the authors data is clearly not the case. Why?
3. More introduction about the perovskite type materials should be added to give the readers a bigger picture. Those includes but not limited to briefly introduce what are perovskite, the advantages, and applications (photocatalysis, etc.). The following paper about perovskite applications must be cited somewhere appropriate ( https://doi.org/10.1021/acsmacrolett.0c00232; https://doi.org/10.1021/jacs.7b00489; https://doi.org/10.1021/acsmaterialslett.1c00785).
4. From the XRD data, there are a lot of peaks that are not perovskite peaks with high doping ratio. Are those impurities. If so, why do not authors remove them? Would them influence the biostudy?
5. Figure 4b, the perovskite seems stable even in water, which is abnormal. Can also provide the characterization of the materials after water stability testing?
6. Figure 1a, I suggest authors give the absorption data instead if it is possible.
Author Response
We would like to thank the comments and suggestions of the reviewer which are definitely helpful for improving the quality of our manuscript. The responses (abbreviated as R) to these comments (abbreviated as C) and the changes made in the revised manuscript (marked in red color) are described in detail in the following.
The authors reported Ce doped Cs3Bi2Cl9 quantum dots for upconversion fluorescence bioimaging. Halide-based perovskite is a hot area and this paper could be attractive to many readers. This work is well organized and performed. However, there still some flaws in the manuscript, including references and writing. Therefore, a major revision must be made before I can reconsider it for publication.
C1: Many typos in the manuscript in the paper and Figures. For example, line 171 “235 cm-1, 262 cm-1 and 284 cm-1”; Figure 2c, I do not know what the B and C mean; Figure 2a and 2b, the number on the scale bar should be the same size.
R1: According to the comment of the reviewer, the text errors in the text have been corrected, and the legend in Figure 2c has also been defined. The scale bar in Figure 2a and 2b have also been unified.
C2: Figure 2c, in principle, the particle size distribution should be Gaussian distribution, while the authors data is clearly not the case. Why?
R2: Thanks for the reviewer's comment. We agree with the reviewer on particle size distribution. Since the particle size in this work is small. the average diameter of particles is between 2-3 nm. In the initial statistics of particle size, particles with diameter less than 2 nm are difficult to be recognized by the statistical software, so the statistical results in Fig. 2c are not comprehensive and small particles are not counted. We add the new TEM images in Figure S4 , and carefully counted the size of particles again, and the new statistical chart obtained has replaced the original one.
C3: More introduction about the perovskite type materials should be added to give the readers a bigger picture. Those includes but not limited to briefly introduce what are perovskite, the advantages, and applications (photocatalysis, etc.). The following paper about perovskite applications must be cited somewhere appropriate ( https://doi.org/10.1021/acsmacrolett.0c00232; https://doi.org/10.1021/jacs.7b00489; https://doi.org/10.1021/acsmaterialslett.1c00785).
R3: According to the suggestion of the reviewer, the references have been added as[2][3][4]
C4: From the XRD data, there are a lot of peaks that are not perovskite peaks with high doping ratio. Are those impurities. If so, why do not authors remove them? Would them influence the biostudy?
R4: Thanks the reviewer for the comments. The alcohol-washed nanocrystal powder is used in the XRD test, and these miscellaneous peaks may come from BiCl3 remaining in the nanocrystal due to excess reactants or other Bi-based halide perovskites generated in the reaction. Due to the synthetic method, these impurities are extremely difficult to remove and the content is not too high, so no further purification. The size of nanocrystals affects cell internalization, and cell internalization influence the biostudy. Most of these impurities are not nano-scale particles and their content is small, which basically does not affect biological performance.
C5: Figure 4b, the perovskite seems stable even in water, which is abnormal. Can also provide the characterization of the materials after water stability testing?
R5: Thank the reviewer for the comment. When synthesizing samples, we refer to the idea of reference [1] and obtain the water stability results of quantum dots similar to those in reference. Water molecules react with BiCl3, and the generated BiOCl coats the surface of the quantum dots to prevent the perovskite from being damaged. Due to the surface pas-sivation of BiOCl, the samples have good water stability.
Figure1:fluorescence spectrum of (a) and Cs3Bi2Cl9 QDs (b)Cs3Bi2Cl9: Ce 30% QDs in different time after adding deionized water.
[1] Ma, Z-Z.; Shi, Z-F.; Wang,L-T.; Zhang,F.; Wu,D.; Yang, D-W.; Chen, X.; Zhang,Y.; Shan ,C-X.; Lia, X-J. Water-induced fluorescence enhancement of lead-free cesium bismuth halide quantum dots by 130% for stable white light-emitting devices. Nanoscale. 2020, 12, 3637–3645
C6: Figure 1a, I suggest authors give the absorption data instead if it is possible.
R6: According to the suggestion of the reviewer, Figure 1a have been change to the absorption spectrum.

Reviewer 3 Report
The authors provide a nice study on Bi-based lead-free perovskite materials and its image for bioimaging applications. The Ce doped Cs3Bi2Cl9 demonstrates up to 22.12% PLQY and 71% enhancement compared to undoped control. This result is well-supported. The reviewer suggest add more discussion on the toxicity on Cs3Bi2Cl9:Ce before acceptance.
Author Response
We would like to thank the comments and suggestions of the reviewer which are definitely helpful for improving the quality of our manuscript. The responses (abbreviated as R) to these comments (abbreviated as C) and the changes made in the revised manuscript (marked in red color) are described in detail in the following.
C1: The authors provide a nice study on Bi-based lead-free perovskite materials and its image for bioimaging applications. The Ce doped Cs3Bi2Cl9 demonstrates up to 22.12% PLQY and 71% enhancement compared to undoped control. This result is well-supported. The reviewer suggest add more discussion on the toxicity on Cs3Bi2Cl9:Ce before acceptance.
R1: Thanks the reviewer for the comments. According to the suggestion of the reviewer, we change the sentences: “When the concentrations of Cs3Bi2Cl9 and Cs3Bi2Cl9: Ce 30% QDs were both 85 μg/mL, the viability of HepG2 cells was 90% and 90%, respectively, especially when the concentration of both was as high as 200 μg/mL, the cell viability was 90% and 90%, respectively.” to “When the concentrations of Cs3Bi2Cl9 and Cs3Bi2Cl9: Ce 30% QDs are low to 10 μg/mL, the viability of HepG2 cells are high to 95.2% and 98.2%, respectively. When the concentrations increase to 40 μg/mL, the viability of Cs3Bi2Cl9 QDs group decrease to 92%,while Cs3Bi2Cl9: Ce 30% QDs group is 96.1%. With the increase of concentration, the data gap between the two experimental groups widened. when the concentration is as high as 200 μg/mL, the viability of Cs3Bi2Cl9 QDs group is 81.3%, that for Cs3Bi2Cl9 QDs group is 91%. Doping Ce not only improves the PLQY, but also reduces the toxicity of quantum dots.”
Round 2
Reviewer 2 Report
The revised version is very good and should be published